# Let's Stop Bleeding! Precise Bleeding Data Estimation and Visualization Methods for Laparoscopic Surgeries

## Abstract

Intraoperative bleeding remains a significant challenge in modern surgery, necessitating rapid and accurate localization of bleeding sources to ensure effective hemostasis. Proactive detection and timely intervention are critical for minimizing blood loss, reducing operative time, preventing complications, and decreasing the need for intensive postoperative care. In this research, we introduce Selective Bleeding Alert Map (SBAM), a novel GAN-based framework designed for precise real-time detection of bleeding origins during surgery. Building upon our earlier BAM framework, SBAM shifts from broad, area-wide alerts to a focused approach that highlights only the exact bleeding areas, enhancing visual accuracy and potentially improving surgeon focus and visibility—particularly beneficial in cases of minor bleeding where excessive alerts could interfere with the surgical process. To achieve this, we developed advanced image-to-image translation and segmentation models, custom thresholding techniques, and trajectory detection algorithms to pinpoint bleeding sources with high precision. Utilizing our developed mimic organ system for ethically sourced, realistic datasets—alongside synthetic data generated from the orGAN system and Large Mask Inpainting (LaMa)—we created a dedicated dataset specifically for SBAM training, including over 1,000 manually annotated images capturing both bleeding and non-bleeding regions within marked bleeding areas. Our instance segmentation model achieved a precision of 92.5%, an accuracy of 98% and a mask mean Average Precision of 85% at an IoU threshold of 0.5 (mAP@50). Additionally, the SBAM model demonstrated high accuracy in detecting bleeding points within real surgical videos from the Hamlyn dataset, underscoring its potential for practical surgical applications. Powered by core algorithms and uniquely developed datasets, SBAM represents a pivotal advancement in AI-assisted surgery, demonstrating superior performance in detecting bleeding regions with high precision during critical scenarios.

## 1 Introduction

Surgery is a delicate field where even minor mishaps can lead to major complications. Despite advancements aimed at reducing operative risks, challenges remain, especially in laparoscopic and minimally invasive procedures, which are hindered by limited visibility and reduced tactile feedback. Rapid and accurate responses to intraoperative issues are essential to prevent long-term damage and the need for extensive postoperative care. One of the primary challenges during any surgical procedure is internal bleeding. The prompt and precise detection of bleeding sources is critical to prevent excessive blood loss, decrease operative time, and minimize postoperative complications. Despite improvements in surgical techniques and technologies, bleeding remains a leading cause of surgical morbidity and mortality.

Globally, surgical complications contribute substantially to the burden of disease. The World Health Organization estimates that complications occur in up to 25% of patients undergoing major surgical procedures, with bleeding being a significant contributor (Weiser et al., 2008). In laparoscopic surgeries, the incidence of bleeding complications, while variable, is noteworthy. For example, bleeding complications during laparoscopic cholecystectomy occur in approximately 2% of cases

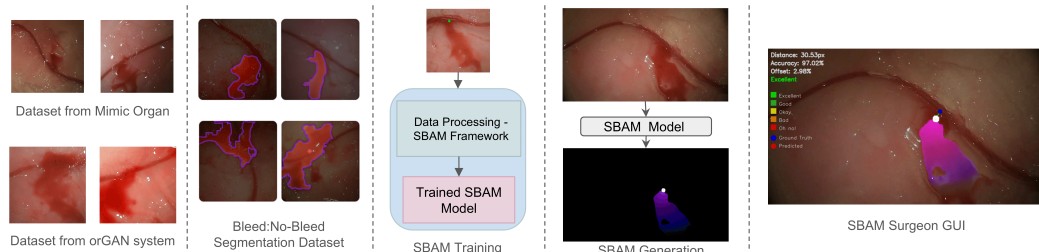

Figure 1: Workflow Overview: Tracing the development from synthetic data generation and processing using Mimic Organ & orGAN system, through model training, to the deployment of the SBAM graphical user interface in surgical settings

(Z'graggen et al., 1998), and intraoperative bleeding in laparoscopic colorectal surgery can occur in up to 6% of procedures (Reissman et al., 1996).

Bleeding often occurs at critical moments during surgery and can quickly obscure the surgical field. Surgeons may initially be focused on other regions, delaying the detection of bleeding until it becomes more serious. Identifying the exact source of bleeding is essential for effective hemostasis but can be difficult to achieve. In laparoscopic surgery, this challenge is exacerbated due to reliance on video cameras that provide a limited field of view. The presence of blood can further obscure the surgical site, complicating the identification of bleeding origins (Abbitt et al., 2017). The lack of tactile feedback inherent in minimally invasive surgery deprives surgeons of an important sensory modality used in open surgeries to detect bleeding (Bholat et al., 1999; Amirabdollahian et al., 2018). Additionally, the dynamic nature of the surgical environment, with the constant movement of instruments and tissues, complicates the differentiation between bleeding and other visual artifacts (Allan et al., 2019).

Delayed control of bleeding can lead to increased transfusion requirements, postoperative anaemia, infection, prolonged hospital stays, and increased mortality (Spahn et al., 2012). Studies have demonstrated that intraoperative blood loss is independently associated with increased postoperative complications and mortality in surgical patients (Wu et al., 2012). Enhancing the ability of surgeons to detect and manage bleeding promptly is therefore essential for improving surgical outcomes. Traditional bleeding detection relies heavily on the surgeon's visual assessment and experience, making it subjective and susceptible to fatigue or distractions (García-Martínez et al., 2017). These methods can be time-consuming and may not offer the rapid detection required in surgery. While automated bleeding detection systems have been developed, many depend on simple thresholding or color segmentation, which lack robustness against variations in lighting, tissue properties, and the presence of other fluids, often resulting in inaccuracies (Mohebbian et al., 2021).

Advancements in Artificial Intelligence (AI) and Machine Learning (ML) offer new possibilities for enhancing bleeding detection during surgery. Deep learning techniques, such as generative adversarial networks (GANs), have shown great promise in image recognition and segmentation tasks in medical imaging (Sorin et al., 2020). Previous studies have applied AI to tasks such as surgical instrument detection in Zhao et al. (2019), anatomical structure segmentation in Toro et al. (2016), and assessment of surgical skill (Pedrett et al., 2022). However, the application of AI for real-time bleeding detection in surgery remains an emerging area of research.

One such AI-based approach is the Bleeding Alert Map (BAM) framework (Sogabe et al., 2023), which enhanced surgical safety by visualizing potential bleeding areas and estimating bleeding start points. While effective for significant bleeding, BAM often issued wide-area alerts for safety reasons, which could interfere with surgeon concentration during minor bleeding by creating unnecessary visual clutter and lacking precise intervention guidance.

To address these limitations, we propose the Selective Bleeding Alert Map (SBAM), a framework that presents focused alerts highlighting only the exact bleeding sites and providing precise localization of the bleeding origin. SBAM refines BAM by minimizing the alerted area and pinpointing the bleeding start point, enhancing the surgeon's ability to respond quickly without distractions—particularly crucial in minor bleeding where excessive alerts are counterproductive.

The key differences between BAM and SBAM lie in both algorithm and function. BAM provided broad alerts based on general bleeding detection without distinguishing bleeding severity or offering precise localization, often lacking specificity on where the surgeon should intervene. In contrast, SBAM employs advanced image-to-image translation, segmentation models, custom thresholding techniques, and trajectory detection algorithms to accurately detect and localize bleeding sources. By focusing on precise bleeding source detection rather than broad area alerts, SBAM enhances visual accuracy and potentially improves surgeon focus and visibility.

To develop SBAM, we utilized a combination of advanced computer vision and AI techniques. We created realistic, ethically sourced images using mimic organ systems (Sogabe et al., 2023) and generated synthetic data with the orGAN system enhanced by LaMa-inspired inpainting. These components enabled us to build a comprehensive dataset for training and validation, ensuring the system can distinguish bleeding sources with high precision even under challenging conditions. The integration of SBAM into surgical practice has the potential to significantly improve patient outcomes by enabling faster hemostasis and reducing the risks associated with delayed bleeding management (Lamb et al., 2023). By enhancing the accuracy of bleeding detection and minimizing unnecessary visual distractions, SBAM aims to set new standards in AI-assisted surgery.

## 2 Data Acquisition

The effectiveness of the SBAM framework hinges on the availability of high-quality, annotated datasets that accurately represent bleeding scenarios in surgical environments.

### 2.1 Development of Ethical Bleeding Datasets

Developing an ethical and diverse bleeding dataset is crucial for SBAM's ability to generalize in real-world surgical scenarios. The dataset comprises two key components: real-world mimicking organ data and synthetic images generated by the orGAN system. Mimic organs are synthetically developed to replicate the appearance, texture, and functionalities of living organs, including bleeding behaviours. The orGAN system complements this by generating high-quality synthetic images. This dual approach ensures the dataset covers a broad spectrum of bleeding scenarios while addressing ethical concerns in medical data collection.

1. **Mimicking Organ Data:** We utilized approximately 200 high-definition videos from our mimicking organ setups, which simulate realistic surgical bleeding scenarios on artificial organs crafted from materials mimicking human tissue properties. Images were taken with a handheld USB microscope and captured in a $1280 \times 720$ RGB, MPEG4 format video (20 fps). Leica CLS 150X (Leica Microsystems, Wetzlar, Germany) was employed as the light source, and the surroundings were covered with a blackout curtain at the time of imaging to reproduce the environment similar to that of an abdominal cavity These setups allow for controlled replication of various bleeding patterns, flow rates, and lighting conditions, providing valuable data for model training. Each video includes a start frame ($F_s$) and an end frame ($F_e$), essential for extracting temporal information and performing the "Pixel Differentiation" step during data preprocessing.

2. **Synthetic Image Generation:** To enhance dataset diversity and address the scarcity of annotated surgical images, we employed the orGAN system to generate around 15,000 high-fidelity synthetic images with non-static temporal information. The orGAN system uses Generative Adversarial Networks (GANs) to produce realistic surgical images featuring a wide range of bleeding scenarios. In-detail explanation of the generation has been mentioned in our previous paper [Paper citation withheld for blind review]. Synthetic data augmentation has been shown to improve model performance by providing variability and aiding generalization in medical imaging tasks.(Frid-Adar et al., 2018).

Both the mimicking organ data and synthetic datasets have precise coordinates for the areas where bleeding originates. However, the synthetic images generated by the orGAN system lack static temporal information (no start and end frames), making pixel differentiation infeasible for these images. Static Temporal Information helps during the "Pixel Differentiation" process in Preprocessing.

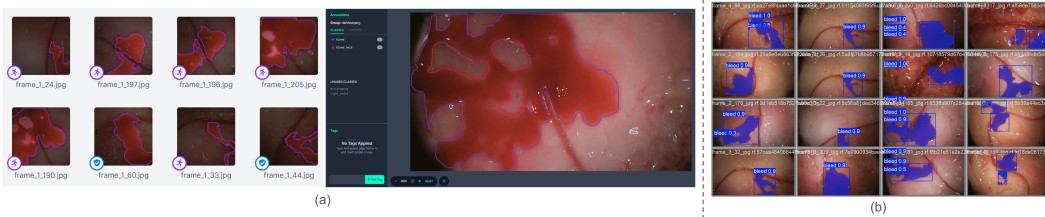

(a)                                                         (b)

Figure 2: (a) Development of the Manually Annotated Dataset for Instance Segmentation of Precise "Bleed-Only" regions (b) Detection of Bleeding Regions by Feature Trained SOTA Model

## 2.2 Limitations of Traditional Computer Vision Techniques

Previous research has often relied on basic computer vision (CV) techniques, such as binary thresholding or color-based bleeding detection (Yuan et al., 2016). Some recent approaches employ classifiers to determine the presence of blood in images or use feature vectors and machine learning models to detect approximate bounding boxes of bleeding areas (García-Martínez et al., 2017). However, these methods struggle in complex surgical environments due to variable lighting conditions, organ colors that resemble blood, and the inherent complexity of surgical scenarios, resulting in inaccuracies such as false positives or missed detections (Moccia et al., 2018).

## 2.3 Development of a Beeding Segmentation Dataset

Recognizing the need for precise bleeding localization (Biancari et al., 2017) and the limitations of existing datasets, we created a first-of-its-kind segmentation dataset comprising over 1,000 manually annotated images from the mimicking organ videos. Over a span of 40 days, expert annotators meticulously labelled each image. Figure 2 (a) shows a glimpse of an example set of segmented images along with its precise borders.

The annotations include two classes:

1. **Bleed Zone:** The exact outlines of all areas exhibiting bleeding, providing precise segmentation of bleeding regions. Visible within the purple segment boundary in Fig 2 (a).

2. **No-Bleed Zone within Bleeding Areas:** Areas within the bleed zones that are not actively bleeding. This class is distinct from regular tissue areas because it highlights regions within the bleeding zone that are unaffected, helping to subtract out areas not affected by bleeding. This differentiation is crucial because standard segmentation may encompass entire regions, and distinguishing non-bleeding areas within the bleeding zone enhances the precision of the model. It is visible as a red segment in Fig. 2 (a).

To isolate the areas actively exhibiting bleeding, we compute the difference between the bleed zone segmentation mask $B(x, y)$ and the no-bleed mask $N(x, y)$. The "Bleed-Only" region is defined as $\{(x, y) \mid B(x, y) - N(x, y) > 0\}$, where the set of pixels of the difference is positive. This operation ensures that only the actively bleeding pixels are retained by subtracting non-bleeding areas within the bleeding zones. The "Bleed" and "No-Bleed" zones can be seen in Fig. 2 (a). The dataset is further divided into 3 splits of - train, test valid in an 8:1:1 ratio in a balanced manner. This detailed segmentation lays the foundation to allow us to train a state-of-the-art (SOTA) model capable of accurately detecting and segmenting specific bleeding-only regions, even in complex surgical scenarios where traditional computer vision techniques and previous works fall short.

## 3 Data Preprocessing

To prepare the dataset for training the Synthetic Bleeding Alert Map (SBAM) model, we implemented a comprehensive preprocessing pipeline that handles both video data (from the mimicking organ dataset) and static images (from the synthetic dataset). The acquired data is required to be processed through multiple steps to produce a training-ready dataset that will enable effective SBAM model training. This preprocessing, as beautifully laid out in Fig. 3, involves several key steps, in-

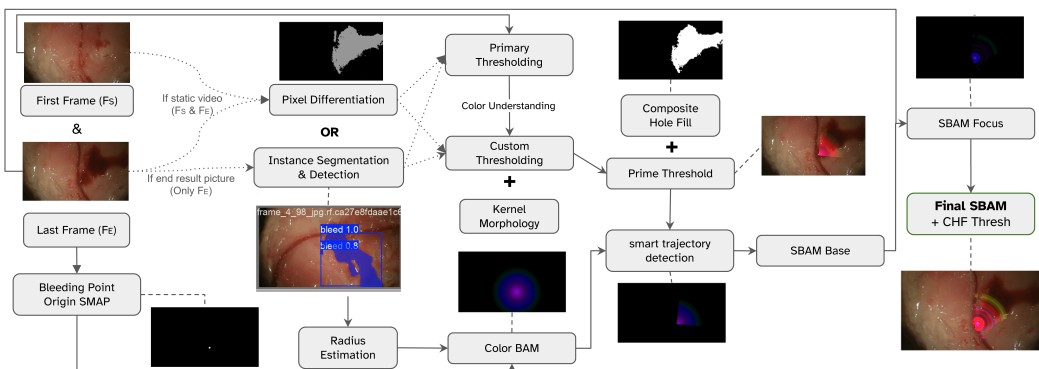

Figure 3: Data Pre-processing Workflow: CV Algorithms & Functions of SBAM

cluding frame extraction, pixel differentiation, segmentation, thresholding, contour selection, morphological transformations, and the generation of the SBAM base with radial highlights.

## 3.1 INITIAL PROCESSING BASED ON INPUT TYPE

The first step depends on whether the input is a video or a static image:

- **Video Input**: For videos, we extract both the first frame $F_s$ and the last frame $F_e$ from each video sequence $V$. These frames are crucial for performing pixel differentiation to detect changes indicative of bleeding: $F_s = \text{get\_first\_frame}(V), \quad F_e = \text{get\_last\_frame}(V)$.

- **Static Image Input**: If both frames are not available (i.e., static images), we use a trained instance segmentation model (YOLOv8), an upgraded version of Redmon et al. (2016) to detect bleeding regions directly: $M_{\text{bleed}}, M_{\text{bleed\_bald}} = \text{YOLOv8}(I)$, where $I$ is the input image, $M_{\text{bleed}}$ is the bleeding mask, and $M_{\text{bleed\_bald}}$ is the non-bleeding area within bleeding zones.

## 3.2 GROSS BLEEDING DETECTION

- **Pixel Differentiation**

  When both $F_s$ and $F_e$ are available, we perform pixel differentiation to detect bleeding regions. We compute the absolute difference between the grayscale versions of the frames: $D(x, y) = |G_s(x, y) - G_e(x, y)|$, where $G_s$ and $G_e$ are the grayscale versions of $F_s$ and $F_e$, respectively. We then apply a threshold $T_{\text{diff}}$ to create a binary mask $M_{\text{diff}}$:

$$M_{\text{diff}}(x, y) = \begin{cases} 1, & \text{if } D(x, y) > T_{\text{diff}}, \\ 0, & \text{otherwise.} \end{cases} \tag{1}$$

- **One-shot Instance Segmentation**

  For static images or when pixel differentiation is not feasible, we utilize our trained segmentation model for instance segmentation to detect bleeding regions directly. We refine the bleeding mask by subtracting the *bleed_bald* mask from the *bleed* mask:
  $M_{\text{bleed\_refined}} = M_{\text{bleed}} - M_{\text{bleed\_bald}}$.

## 3.3 MASK REFINEMENT AND CONTOUR SELECTION

The detected areas, especially from the pixel differentiation method, can be extremely noisy. A primary thresholding uses morphological operations to focus on the pixels that best match our specific requirements. We apply morphological opening and closing with a structuring element $K$ (e.g., a $3 \times 3$ kernel) to remove noise and fill small holes: $M_{\text{morph}} = \text{morph\_close}\left(\text{morph\_open}(M_{\text{diff}}, K), K\right)$.

To focus on the most relevant bleeding region, we detect contours in the binary mask and select the largest contour nearest to the known bleeding point coordinates $(x_b, y_b)$. For each contour $C_i$, we

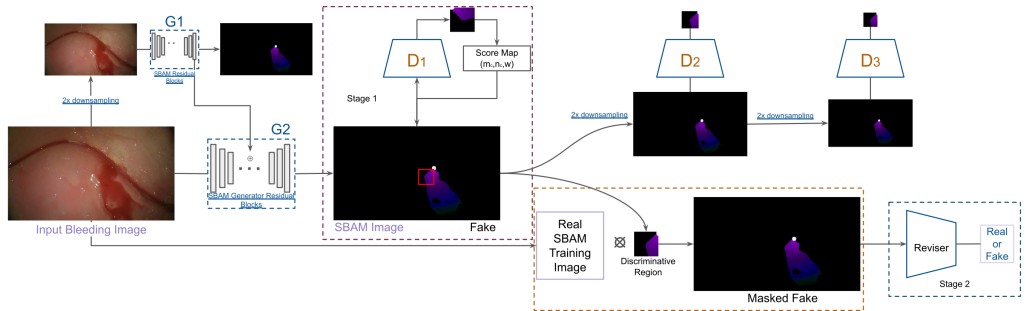

Figure 4: SBAM Image-to-Image Translation Model's Training Architecture

compute its area $A_i$ and centroid $(x_i, y_i)$, and select the contour that minimizes the distance-to-area ratio:

$$C^* = \arg\min_{C_i} \left( \frac{(x_i - x_b)^2 + (y_i - y_b)^2}{A_i} \right). \tag{2}$$

We further refine the selected contour mask $M_{\text{contour}}$ by applying morphological closing to fill composite holes, resulting in a perfect, noise-free solid segmentation: $M_{\text{filled}} = \text{morph\_close}(M_{\text{contour}}, K)$.

### 3.4 BLEEDING ALERT MAP (BAM) CREATION

The bleeding point coordinates $(x_b, y_b)$ are used to create the colour Bleeding Alert Map (BAM). The detected area is estimated for its gross radius relative to the size of the highlight dynamically, allowing us to get a balanced dataset with different sizes. We generate a radial gradient centred at the bleeding point with a maximum radius $R_{\text{max}}$. We define the distance from the bleeding point for each pixel $(x, y)$: $d(x, y) = \sqrt{(x - x_b)^2 + (y - y_b)^2}$.

The highlight image $H$ is generated using a VIBG (Violet-Indigo-Blue-Green) fade function, mapping the normalized distance to a colour gradient:

$$H(x, y) = \begin{cases} \text{VIBG\_fade}\left( \frac{d(x,y)}{R_{\text{max}}} \right), & \text{if } d(x, y) \leq R_{\text{max}}, \\ 0, & \text{otherwise.} \end{cases} \tag{3}$$

The VIBG_fade function interpolates colors based on the distance ratio:

$$\text{VIBG\_fade}(t) = (1 - t) \cdot \text{color}_1 + t \cdot \text{color}_2, \tag{4}$$

### 3.5 SBAM BASE GENERATION AND FOCUS

We combine the BAM with the bleeding mask to get the base SBAM: $M_{\text{SBAM}} = H \odot M_{\text{filled}}$, where $\odot$ denotes element-wise multiplication. To detect the direction of the flow of blood, we analyze the distribution of bleeding pixels relative to the bleeding point. We compute the angle $\theta$ for each bleeding pixel: $\theta_k = \arctan 2(y_k - y_b, x_k - x_b)$, and calculate the mean angle: $\theta_{\text{mean}} = \frac{1}{N} \sum_{k=1}^{N} \theta_k$.

We define a segment mask $S(x, y)$ to focus the SBAM as the below equation where $\Delta\theta$ is the angular width of the highlight segment. The final SBAM highlight is: $H_{\text{final}}(x, y) = H(x, y) \cdot S(x, y)$.

$$S(x, y) = \begin{cases} 1, & |\arctan 2(y - y_b, x - x_b) - \theta_{\text{mean}}| \leq \frac{\Delta\theta}{2}, \\ 0, & \text{otherwise,} \end{cases} \tag{5}$$

We overlay the final SBAM onto the original image $I$: $I_{\text{SBAM\_final}} = \alpha I + \beta \left( H_{\text{final}} \odot M_{\text{filled}} \right)$, where $\alpha$ and $\beta$ are blending coefficients set to balance the visibility of the original image and the SBAM.

## 4 MODEL TRAINING

### 4.1 INSTANCE SEGMENTATION MODEL TRAINING

The detection of bleeding in surgical videos was approached as an instance segmentation task, leveraging a YOLOv8-based architecture with specific adaptations for segmentation (Che et al., 2023). The model was fine-tuned on a dataset specifically annotated for bleeding detection, comprising over 1,000 manually annotated images that included both bleeding and non-bleeding regions. Annotations were made for two classes: *bleed* and *bleed_bald* (non-bleeding areas within bleeding zones). To address the challenge of limited annotated data, we employed techniques inspired by one-shot learning Wang et al. (2020), which is advantageous in medical scenarios where annotated data is scarce and costly to obtain. Data augmentation methods, such as random flipping, scaling, and rotation, were applied to enhance the model's robustness to variations in the data.

The bleeding segmentation model was trained using the following hyperparameters: 50 epochs, a batch size of 24, an initial learning rate of 0.01, a weight decay of 0.0005, and an image size of $640 \times 640$ pixels. An adaptive optimizer was used, and pre-trained weights were loaded to facilitate quicker convergence. The model employed overlapped masking with a mask ratio of 4 to improve segmentation quality. The loss function for training combined four components:

$$L_{\text{total}} = \lambda_{\text{box}} L_{\text{box}} + \lambda_{\text{seg}} L_{\text{seg}} + \lambda_{\text{cls}} L_{\text{cls}} + \lambda_{\text{DFL}} L_{\text{DFL}}, \tag{6}$$

where $L_{\text{box}}$ represents the bounding box regression loss, $L_{\text{seg}}$ is the segmentation mask loss, $L_{\text{cls}}$ is the classification loss, and $L_{\text{DFL}}$ is the distributional focal loss. The weights $\lambda_{\text{box}}$, $\lambda_{\text{seg}}$, $\lambda_{\text{cls}}$, and $\lambda_{\text{DFL}}$ were set to balance the contribution of each term, ensuring that the model effectively learned to localize and classify bleeding regions.

The performance metrics confirmed that the model converged effectively with minimal overfitting, supported by the validation loss trends. The combination of Binary Cross-Entropy (BCE) and Dice loss functions in $L_{\text{seg}}$ improved segmentation accuracy by balancing pixel-wise classification and overlap measures:

$$L_{\text{seg}} = \sum_{i=1}^{N} \left( \text{BCE} \left( \hat{y}_i, y_i \right) + \text{Dice} \left( \hat{y}_i, y_i \right) \right), \tag{7}$$

where $N$ is the number of pixels, $\hat{y}_i$ is the predicted label, and $y_i$ is the ground truth label. The model's robustness was evident in its ability to handle varying lighting conditions and complex tissue structures, facilitated by the use of pre-trained weights, overlap masking, and appropriate hyperparameters.

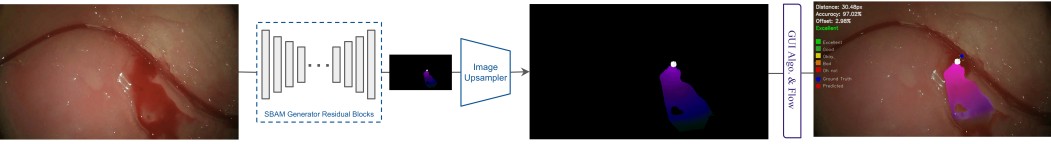

Figure 5: SBAM generated through the Model along with the Surgeon GUI Output

### 4.2 SBAM GENERATION TRAINING

In parallel, SBAM generation was framed as an image-to-image translation task, translating original surgical images into their corresponding SBAM representations (aka, the data produced via Data Preprocessing). For this purpose, we employed the NVIDIA Pix2PixHD Architecture (Wang et al., 2017), which is well-suited for high-resolution image translation tasks, with minimal tweaks and customisations to suit our needs. This architecture, as shown in Fig. 4 enables the direct mapping of raw input images to segmented bleeding regions in a visually interpretable manner.

The architecture utilizes a two-tier generator approach ($G1$ and $G2$) for high-resolution image-to-image translation. $G1$ generates a basic, low-resolution structure from the input, which is defined by $G2$ to enhance detail and quality. The output undergoes evaluation by multi-scale discriminators ($D1$, $D2$, $D3$) at various resolutions to ensure photorealism and accuracy (Kirimanjeshwara & Prasad, 2024). $D1$ produces a score map identifying critical discriminative regions, crucial for tasks like precise bleeding detection in medical imaging, with further refinement by $D2$ and $D3$.

The model was trained using the previously discussed dataset of surgical images, with the SBAM images generated during preprocessing serving as target outputs. The input resolution was set to $1024 \times 1024$ pixels to preserve spatial details crucial for accurate bleeding area mapping. The training was conducted over 200 epochs with a batch size of 2, using an initial learning rate of 0.0002 and the Adam optimizer with $\beta_1 = 0.5$. The loss function used for the Pix2PixHD model combined adversarial loss ($L_{\text{GAN}}$), feature matching loss ($L_{\text{FM}}$), and perceptual loss ($L_{\text{VGG}}$):

$$L_{\text{total}} = L_{\text{GAN}} + \lambda_{\text{FM}} L_{\text{FM}} + \lambda_{\text{VGG}} L_{\text{VGG}}, \tag{8}$$

$$where: L_{\text{GAN}} = \mathbb{E}_y \left[\log D(y)\right] + \mathbb{E}_x \left[\log \left(1 - D(G(x))\right)\right], \tag{9}$$

with $G$ being the generator network, $D$ the discriminator, $x$ the input image, and $y$ the target SBAM image. The feature matching loss $L_{\text{FM}}$ and the perceptual loss $L_{\text{VGG}}$ ensure that the generated images are not only indistinguishable from real images by the discriminator but also similar in feature space and perceptually to the target images. Figure 5 hints at how the trained model is used during inference along with the GUI the surgeon would be greeted with. The SBAM model successfully learned to generate the SBAM output images along with Bleeding Point Origin that closely resembled the ground truth.

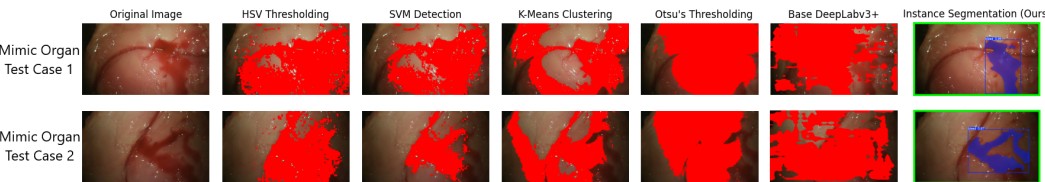

Figure 6: Comparison of Existing Methods to Generate Segmentation Maps for Bleeding Regions vs our model trained on our developed dataset

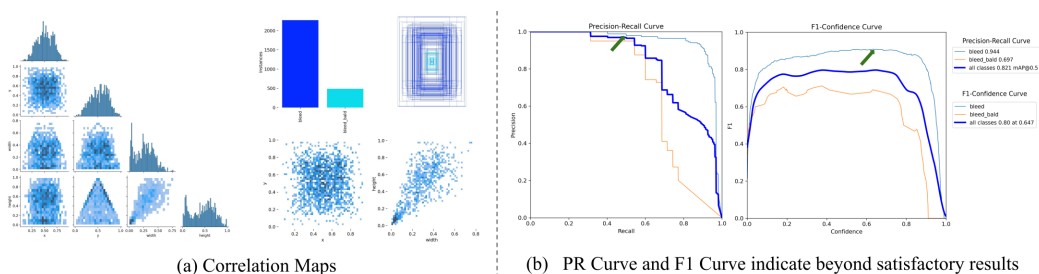

(a) Correlation Maps
(b) PR Curve and F1 Curve indicate beyond satisfactory results

Figure 7: (a) Correlation Maps of the Instance Segmentation Model (b) 0.944 @ Precision-Recall and 0.91 F1 Confidence for Segmented Bleeding Regions

## 5 RESULTS AND DISCUSSIONS

The instance segmentation model trained with our dataset provided exemplary results, setting a record for semantic segmentation for bleeding. This can be attributed to the extremely accurate segmentation dataset and the SOTA architecture. Figure 6 compares our Instance Segmentation Results with the publicly available alternatives online that are a common reference for Bleeding Segmentation in past works. Over the training epochs, the model exhibited significant improvements in key metrics. The bounding box precision increased from 86.6% in the first epoch to 92.5% by the 50th epoch, and the bounding box recall improved from 75.7% to 82.0%. The mask mean Average Precision at an IoU threshold of 0.5 (mAP@50) increased from 82.3% to 85.6%, and the mask mAP@[0.5:0.95] increased from 62.5% to 69.2%. The F1 score, as shown in the plot in Fig. 7 (b) of bleeding regions was 0.95 after training. The loss values decreased consistently, indicating effective learning. The main requirement for us is the accuracy and precision of the bleeding area and in this aspect, the Segmentation regions were noted to be around 98% accurate during Validation. Figure 7 (a) is a handy complication of various correlation maps of the datasets and the results.

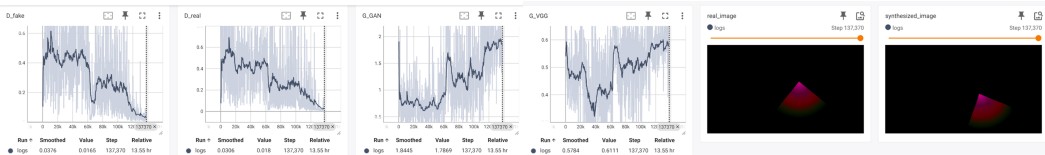

Figure 8: Snippet of Satisfactory improvement in Generator and Discriminator losses for the I2I GAN Network. Although the numbers are satisfactory, GAN is a domain where Visual Confirmation is the most accurate way of confirming model improvement

The primary goal of this research was to train and fine-tune the SBAM architecture (Figure 4) to develop an exemplary SBAM model. Image-to-image translation is particularly challenging due to the sensitivity of results to parameter settings. Throughout the training, we observed a steady decrease in generator losses, indicating effective learning of realistic SBAM images. At epoch 1, iter 100, the adversarial loss was 0.857, feature matching loss was 4.876, and perceptual loss was 1.162. By epoch 128, iter 12,348, these losses improved to 0.417, 2.442, and 0.885, respectively.

To assess the accuracy of the bleeding detection and SBAM generation models during inference, we developed a visualization and evaluation pipeline that overlays predicted bleeding points onto input images along with performance metrics as directed in Fig.5. The evaluation process involves detecting predicted bleeding points, merging close detections to avoid redundancy, calculating spatial discrepancies to the ground truth, and determining an overall accuracy measure. This real-time feedback is crucial for surgeons to understand the reliability of the detections without distractions. Figure 8 shows the positive results during training, supported both via Loss plots and visuals. However, it is worth noticing that, since GANs are domains where quantitative metrics have limited utility, true results are best verified visually (Borji, 2018; Kumar et al., 2023).

We utilise a clustering approach to merge nearby predicted points in SBAM results into a single representative point. Let $\{(x_i, y_i)\}$ be the set of predicted coordinates. We applied a clustering algorithm with a distance threshold $\varepsilon$ to group points and calculated the centroid of each cluster in the below formula where $N_c$ is the number of points in the cluster. This ensures that closely located predictions are consolidated, reducing noise in the evaluation.

$$(x_c, y_c) = \frac{1}{N_c} \sum_{i=1}^{N_c} (x_i, y_i), \tag{10}$$

To quantify the spatial discrepancy between the predicted bleeding point $(x_p, y_p)$ (after merging) and the ground truth coordinate $(x_a, y_a)$, we calculated the normalized Euclidean distance $\mathcal{D}$ relative to the image dimensions as below where $W$ and $H$ are the image width and height, respectively. The normalized distance $\mathcal{D}$ ranges from 0 to 1, representing the proportion of the maximum possible distance within the image.

$$\mathcal{D} = \frac{\sqrt{(x_p - x_a)^2 + (y_p - y_a)^2}}{\sqrt{W^2 + H^2}}, \tag{11}$$

The accuracy $\mathcal{A}$ is then defined as: $\mathcal{A} = (1 - \mathcal{D}) \times 100\%$, where an accuracy of 100% indicates a perfect prediction (zero distance), and lower accuracies correspond to larger discrepancies between the predicted and actual bleeding points. To enhance the model's usability in surgical applications, we developed a visualization and evaluation GUI that overlays predicted bleeding points (red circles) and ground truth points (blue circles) onto input images, accompanied by performance metrics. Indicators such as Euclidean distance, accuracy, and error percentages provide a concise yet comprehensive evaluation, ensuring clarity for real-time surgical decision-making. By consolidating close detections, normalizing distance metrics, and maintaining consistency across varying image resolutions, the pipeline optimizes applicability across diverse surgical contexts.

Figure 9 provides the conclusive validation required to demonstrate the model's functionality and usability in real-world scenarios. We are able to compare and show the possible effectiveness and clear-cut advantages SBAM provides over regular BAM. The model's robustness was further evaluated using the Hamlyn dataset (Mountney et al., 2010; Stoyanov et al., 2010; Pratt et al., 2010), which comprises actual surgical videos of human procedures. Remarkably, despite being trained entirely on synthetic datasets—generated from mimic organs and the orGAN system—our model

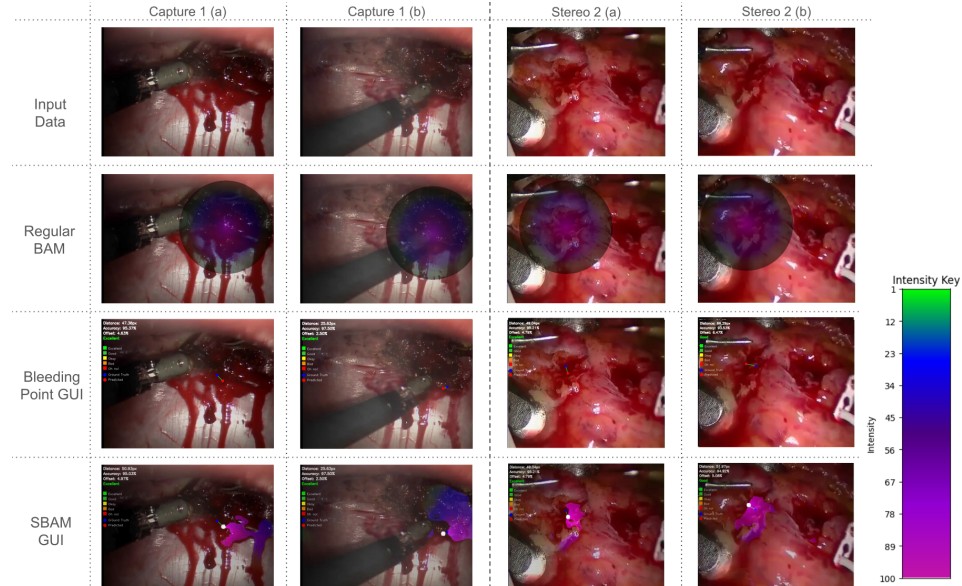

Figure 9: SBAM tested with in-vivo Hamlyn Dataset. The positive results on unseen data is a validation for the system's performance and capability

demonstrated high accuracy in detecting bleeding points within these real surgical videos. This achievement underscores the model's ability to generalize from synthetic training data to complex, real-world surgical scenarios, which is pivotal for future advancements in AI-driven surgery. This validation not only highlights the quality of the synthetic data but also its capacity to overcome the practical and ethical constraints of acquiring large-scale annotated surgical data from human subjects (Gao et al., 2022; Satapathy et al., 2023; Li et al., 2021; Collins et al., 2021).

## 6 LIMITATIONS AND CONCLUSION

While the SBAM framework demonstrates significant advancements in precise bleeding detection, certain limitations must be acknowledged. Firstly, we were unable to directly compare our segmentation results with some previous AI-based approaches that utilize bounding boxes and convolutional neural networks (CNNs), due to the unavailability of their datasets and model implementations, which are often proprietary and not publicly shared (Tantoso et al., 2019; Mishra et al., 2022). This lack of accessible benchmarks limits the ability to quantitatively assess the relative performance of our approach. However, we believe our model offers superior performance by providing precise segmentation rather than approximate localization. Other works often rely on bounding boxes or less accurate methods and do not employ advanced techniques like one-shot instance segmentation, as shown in Fig. 6.

Secondly, our training dataset primarily represents classic cases of bleeding on standard organ models, which may not encompass the full spectrum of anatomical variations and complex surgical scenarios encountered in practice. Addressing this limitation will require the development of additional mimic organs and enhancements to the orGAN system to generate more diverse datasets—a focus of our ongoing research.

SBAM offers an effective approach to real-time bleeding source detection, enhancing precision without obstructing critical visual information. Despite the limitations, our work lays a solid foundation for further advancements, and we are committed to expanding our datasets and refining our models to encompass a wider range of surgical conditions.

### DATASET AND TRAINING CODE

The dataset, model weights, training code, and detailed instructions for replication will be released upon the publication of this paper and the completion of subsequent research.

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

# APPENDIX

## A    DATASET AND ETHICAL CONSIDERATIONS

Developing AI models for surgical applications requires extensive, high-quality datasets. However, acquiring real surgical data poses significant ethical and practical challenges due to strict privacy laws like the Health Insurance Portability and Accountability Act (HIPAA) and the General Data Protection Regulation (GDPR), which protect patient confidentiality and limit data availability (Price & Cohen, 2019). Ethical concerns surrounding patient consent and the invasive nature of data collection further complicate the gathering of diverse datasets necessary for robust AI training(Morley et al., 2020).

Traditional reliance on animal models raises additional ethical issues and often fails to accurately replicate human anatomical and physiological conditions, limiting the effectiveness of AI models trained on such data (Akhtar, 2015). To overcome these challenges, we developed an innovative approach using synthetic data generated from in-house mimic organ system.

### A.1    ETHICAL INNOVATION WITH SYNTHETIC DATA

Mimic organ system creates realistic organ models using biocompatible materials that simulate human tissue properties. This allows us to replicate surgical scenarios, including internal bleeding, without involving real patients or animals. By simulating various bleeding patterns and surgical conditions, we generate high-fidelity surgical images ethically and efficiently. We further augmented our dataset using Generative Adversarial Networks (GANs). As noted before, GANs have been effectively utilized to generate synthetic medical images that enhance AI training while preserving patient privacy (Frid-Adar et al., 2018). By training GANs on existing surgical images, we produced a diverse set of synthetic images capturing various surgical anomalies and conditions. This method allows us to explore a wide range of surgical scenarios, ensuring our model is trained on comprehensive data while adhering to ethical standards.

Accurate annotation of surgical images is essential but often time-consuming and requires expert knowledge. By using synthetic data with known ground truth, we simplified the annotation process. The exact locations and characteristics of bleeding points are inherently known in our synthetic images, reducing the burden on medical professionals for manual annotation. Expert reviews and validation on these annotations were done to ensure high-quality labels necessary for effective AI training. Our dataset comprises over 1,000 images, including both real and synthetic data, covering various surgical procedures, bleeding types, and environmental conditions. By incorporating diverse blood flow patterns, organ appearances, and lighting conditions, we enhanced the model's ability to generalize across different clinical scenarios. This approach addresses the issue of data imbalance often found in medical datasets and improves the robustness and accuracy of the SBAM framework when deployed in real-world settings.

