# OpenReview forum: "Let’s Stop Bleeding! Precise Bleeding Data Estimation & Visualization Methods for Laparoscopic Surgeries"
_ICLR.cc/2025/Conference — ICLR 2025 Conference Withdrawn Submission_

### Official Review · Reviewer_7AtK · 2024-10-27

**Soundness:** 2
**Presentation:** 2
**Contribution:** 3
**Rating:** 5
**Confidence:** 3

**Summary:**

The paper presents a methodology for instance segmentation in surgical image analysis to segment the active bleeding area. Their proposed model reaches a high performance with the help of using a fine-tuned YOLOv8 model.

**Strengths:**

1. The idea of implementing an instance segmentation model on detecting active bleeding spots is interesting.

2. The use of the pixel differentiation to analyze/detect active bleeding area is helpful.

3. Developed a bleeding segmentation dataset.

**Weaknesses:**

1. It would be helpful to include ablation study to analysis the loss weights since there are multiple losses used in this study. Particularly,  the impact is using the adversarial loss (LGAN), feature matching loss (LFM), and perceptual loss (LVGG) needed to be further investigated and analyzed to demonstrate each loss function's effectiveness in solving this task.

2. The figures in Figure 7 is too small, and it is hard to visualize. The authors can take some actions to help with improving the readability, such as increasing the figure size, splitting it into multiple figures, or highlighting key areas that need better visibility.

3. There is lacking quantitative evaluations. I understand the concerns stated in the limitations; However, it might be helpful to compare using different DL models to fine tune, such as the demonstrating the performance using other popular segmentation models as an ablation study to justify why selecting the YoloV8 model.

**Questions:**

1. YoloV8 is a model that released in 2023, why not trying to use the latest release YoloV9?

2. I am also curious whether occlusion of the non-bleeding areas by surgical tools or the surgeon's hands could impact the accuracy of segmenting the active bleeding region, as it may interfere with the pixel differentiation calculations.

---

### Official Review · Reviewer_gGd7 · 2024-10-30

**Soundness:** 2
**Presentation:** 3
**Contribution:** 2
**Rating:** 3
**Confidence:** 3

**Summary:**

A novel framework, Selective Bleeding Alert Map (SBAM), which uses GAN-based image segmentation to focus on precise bleeding detection, aiming to reduce unnecessary distractions and improve visualization accuracy in laparoscopic surgeries. The SBAM framework is supported by a synthetic dataset of annotated surgical images, offering a new method for enhancing bleeding detection accuracy during operations.

**Strengths:**

SBAM addresses an unmet need in surgical procedures by improving visual focus and response to bleeding, which is critical in minimally invasive surgeries. The model achieved strong performance metrics (precision and accuracy), indicating high potential for real-world surgical integration. The use of a synthetic dataset for model training reduces the reliance on real surgical images and associated privacy concerns.

**Weaknesses:**

The model has limited applicability to other (non-)surgical contexts, aims at providing a specialized engineered solution to an application.

The reliance on synthetic data might limit the model’s robustness in diverse real-world surgical scenarios. Validation on a broader set of actual surgical videos or images would enhance the results’ credibility.

The framework’s complexity could pose challenges in practical surgical scenarios.

The paper does not provide an extensive exploration of ablation studies or comparisons with a broad range of baseline methods. The evaluation mainly focuses on demonstrating SBAM’s performance. there is no mention of systematic ablations that examine the impact of individual components (e.g., GAN-based architecture, segmentation thresholds) or a detailed comparison with established detection methods. comparisons with baseline methods or additional ablations could better establish SBAM’s unique contributions and clarify how each component of the model contributes to its overall performance.

**Questions:**

How does the model perform on real surgical data beyond the Hamlyn dataset, and are there plans for further validation?

Could the approach be generalized to other forms of bleeding detection, or is it specific to this surgical type?

Have you performed ablation studies to evaluate the impact of individual components within the SBAM framework, such as GAN-based segmentation or specific thresholding techniques?

How does SBAM compare to other baseline methods in bleeding detection, and could these comparisons be included to strengthen the evaluation?

---

### Official Review · Reviewer_JV1y · 2024-10-31

**Soundness:** 2
**Presentation:** 2
**Contribution:** 2
**Rating:** 3
**Confidence:** 5

**Summary:**

The detection of bleeding zones/regions is a crucial step during surgical procedures. A system indicating the bleeding regions within the surgical scene to the surgeon essentially helps in improving the patient outcome. This work proposes a GAN-based bleeding detection method for precise localization of the bleeding region. Synthetic datasets are designed to train this method and the work claims to improve the instance detection of bleeding zones in real surgical scenes.

**Strengths:**

- This work addresses a crucial yet underexplored issue of bleeding detection during surgical procedures. IA GUI framework is developed to make the detection tool easily accessible for surgeons.
- This work defines a synthetic dataset generation pipeline to create a labeled bleeding dataset. This dataset would be a valuable contribution to the surgical community if the authors make it publicly available.

**Weaknesses:**

- Contribution of the work: The contribution of this work is unclear (see questions)

- Details regarding the dataset are missing (see questions)

- Robustness to real-surgical environments: This work claims that previous methods fail to detect blood regions in real surgical environments under varying lighting conditions or when the organ is the same color as the blood (lines 178-179). However, these claims are not tested using the proposed methods. Although experiments were conducted on the Hamlyn dataset, only qualitative results are presented. This raises the question of who evaluated these qualitative results. For intricate applications such as bleeding detection, qualitative evaluation by doctors or surgeons is essential, alongside quantitative comparisons, to show improved performance.

- Evaluation of variations in bleeding patterns: This work claims to use a mimic organ method to develop a dataset of bleeding events with variations in flow, patterns, and lighting conditions (lines 140-149). However, these variations are not evaluated. A small subset of real bleeding events with such variations should be collected, and the dataset from the mimic organ setup should be compared to validate these claims. For qualitative comparison, a user study involving doctors is necessary, and for quantitative comparison, a simple bleeding detection (classification) model—classifying whether bleeding is occurring or not—can be used. This step is crucial to validate the robustness of the mimic organ model.

- Lack of baselines and ablation: This work lacks a comparison to baselines or ablations (see questions)

**Questions:**

- Contribution:
Does this work propose a method to generate a synthetic bleeding dataset? If so, what real dataset was used to construct the synthetic dataset? What anatomies were focused on in this study, and why was this important for defining the bleeding region? Line 185 mentions 1,000 annotated images—could the authors clarify why this number was chosen when, with a synthetic data generation system, the number of images could be scaled beyond that?
Is a method being proposed to detect bleeding regions in a given surgical scene? This clarification is needed to judge the contribution of this work.

- Overlap in data: Clarification on the dataset used for training and testing.
1.  What was the real dataset used to generate synthetic bleeding videos using the mimic organ system?
2. What was the dataset used to train the orGAN system to generate synthetic bleeding images ?
3.What was the dataset the YOLO detection algorithm was trained on ?
4. What was the size of all the split dataset used for each of these training runs ? The work mentioned only a dataset size of 1000 annotated images, which was from the mimic system.
The YOLO instance segmentation was trained on these 1000 images. Could the authors clarify the size and nature of the test dataset ?

- The need for image-to-image (I2I) translation: Pix2PixHD [i] is a paired image-to-image translation approach, where the input is a bleeding image and the output is a bleeding alert map highlighting the exact bleeding region. Why was this task framed as an I2I translation instead of a simple segmentation task? More clarification is needed to understand the importance of using I2I in this work.

- Is there overlap in the datasets between the training of these modules? In line 425, the instance segmentation method achieves a precision of 86.6% in the first epoch. Does this suggest a data leak between the generation and segmentation training? If not, could the authors provide the training and validation curves and clarify why the score is so high in the first epoch?

- Baselines: Why was the proposed method (SBAM) not compared against the BAM[ii] method which served as the base framework for this work ? The BAM method gives a wider area of bleeding from which an instance segmentation mask can be constructed using the same YOLOV8 model in section 3.1.
Why was Pix2PixHD chosen ? There are other improved versions such as SPADE [iii] (GAN-based), ControlNet [iv], T2I-adapter [v] (diffusion-based) for semantic image synthesis. An ablation of these methods is necessary to show why Pix2PixHD might be the better choice.
The SBAM method is based on image-to-image translation method. There exist two generators, three discriminators and two different faces. What and why was this procedure chosen ? Ablation on each of these components is necessary to show why this specific I2I method was used in the study
What was the ratio of real and synthetic datasets used for training the YOLO segmentation method ? This ablation indicates the need for synthetic images and can show for which type of bleeding types does it improve/decrease performance.

- The figures 7 and 8 are difficult to read. Could the authors explain how a lower generator loss means a better GAN training and high quality images being generated (lines 442-446). How to account for model collapse in this case ?

- In line 521 it is mentioned that prior works do not release their code to compare them against this work. However, I find this contradictory, that this work also mentioned that the data and code would be released upon the completion of a subsequent work. Could the authors clarify their statement?

- What is the inference latency of the deployed model on the GUI. Would this serve in real-time applications ? Could the authors comment on how to improve such a system? This would be necessary when such a system needs to be used by doctors in a real-surgical environment, which this work claims to be suitable for.

Minor suggestions:
Improve the wordings within the figures to make it easier for the reviewer to read the contents of the image (Fig.2,4,5,7,8)
The authors could read through the manuscript and use scientific language for writing. Words such as “beautifully, greeted” could/should be avoided for a scientific paper.

References:
i. TC Wang et al., High-Resolution Image Synthesis and Semantic Manipulation with Conditional GANs
ii. M.Sogabe et al., Bleeding alert map (bam): The identification method of the bleeding source in real organs using datasets made on mimicking organs
iii. T.park et al., Semantic image synthesis with SPADE
iv. L.Zhang et al., Adding Conditional Control to Text-to-Image Diffusion Models
v. Mou et al., T2i-adapter: Learning adapters to dig out more controllable ability for text-to-image diffusion models

**Details Of Ethics Concerns:**

This work mentions the previous framework of the authors in the abstract (line 19-20), the mimic organ system in line 25 and the previous work system in line 141. The mentioned framework is cited in the paper in line 91, 116 and therefore breaks the double anonymity clause.

---

### Official Review · Reviewer_JULy · 2024-11-03

**Soundness:** 2
**Presentation:** 2
**Contribution:** 2
**Rating:** 3
**Confidence:** 3

**Summary:**

The paper introduces a method for detecting and segmenting bleeding regions in surgical images, addressing a need for real-time visualization during procedures. The proposed approach begins with a set of preprocessing steps that create an initial bleeding segmentation mask. Following this, a generative model is applied to perform an image-to-image translation, producing refined segmentation masks from the RGB input images. Given the challenges of limited medical data availability, the model training leverages video data generated from phantom sources and synthetic still images created with the orGAN model.

**Strengths:**

Automatic bleeding detection represents an important task during surgical procedures to avoid complications during surgeries; hence, the paper addresses a relevant problem.

**Weaknesses:**

One of the main limitations of the work is the lack of comparison with recent models. Even though this is mentioned in the limitations section of the paper, it is relevant to include additional models defined for segmentation in the comparison, as this sets a baseline that allows evaluating the performance of the work. In this regard, while they might not be open works for bleeding, there is a group of work on general-purpose medical segmentation, including the UNet, nnUnet, and its recent variants, that can be potentially employed to segment the bleeding regions' still images and in video data.

**Questions:**

Additionally to the limitation presented in the weaknesses section, some questions regarding the methodology are the following:
* How is the preprocessing related to the translation problem? Is the input to the image translation problem the image itself or the output for the preprocessing stage?
* Line 241 mentions there is a model based on the YOLOv8 model to detect bleeding in static images. Is this the same model described in section 4.1?
* In the pixel differentiation stage, it is mentioned that the model employs the first and last frames to detect bleeding as a change in the grayscale images. How sensitive is this stage to non-bleeding-related changes, like tissue ablation?
* Why is the pixel differentiation stage necessary if the YOLO model can give an initial bleeding location?

It is recommended that the font size be increased for the figures, as some illustrations are hard to read.

---

### Note · Authors · 2025-02-10

I have read and agree with the venue's withdrawal policy on behalf of myself and my co-authors.

---

### Meta-Review · Area_Chair_njqD · 2024-12-11

**Metareview:**

The paper presents an approach to detect bleeding region. The reviewers have raised some concerns on the paper, including the limited novelty, unclear contribution, lack of comparison, etc. The authors failed to provide any response to these concerns. Therefore, the paper cannot be accepted for the conference.

**Additional Comments On Reviewer Discussion:**

There is a consensus that the paper shall be rejected and no discussion is needed

---

### Decision · Program_Chairs · 2025-01-22

Reject